# Aposymbiotic Specimen of the Photosynthetic Sea Slug *Elysia crispata*

**DOI:** 10.3390/d14050313

**Published:** 2022-04-20

**Authors:** Paulo Cartaxana, Diana Lopes, Begoña Martinez, Patrícia Martins, Sónia Cruz

**Affiliations:** ECOMARE—Laboratory for Innovation and Sustainability of Marine Biological Resources, CESAM—Centre for Environmental and Marine Studies, Department of Biology, University of Aveiro, 3810-193 Aveiro, Portugal; pcartaxana@ua.pt (P.C.); lopes.diana@ua.pt (D.L.); bego.martinez@ua.pt (B.M.); patriciatmartins@ua.pt (P.M.)

**Keywords:** chloroplast, kleptoplasty, photosynthesis, Sacoglossa

## Abstract

*Elysia crispata* is a sacoglossan sea slug that retains intracellular, functional chloroplasts stolen from their macroalgal food sources. *Elysia crispata* juveniles start feeding on the algae following metamorphosis, engulfing chloroplasts and turning green. In laboratory-reared animals, we report one juvenile “albino” specimen unable to retain chloroplasts. Within 6 weeks post-metamorphosis, the aposymbiotic sea slug was significantly smaller than its chloroplast-bearing siblings. This evidence highlights that chloroplast acquisition is required for the normal development of *E. crispata*.

*Elysia crispata* (Mörch, 1863) (Figure 1) is a sacoglossan sea slug (Gastropoda, Mollusca) common throughout the Caribbean, inhabiting mangrove areas and coral reefs from shallow to 25 m deep waters [1,2]. A striking feature of this sea slug and other relatives of the Sacoglossa order is the capacity to digest the cellular content of their algal food sources while retaining intact functional chloroplasts (kleptoplasts) [3,4]. Kleptoplast-bearing sea slugs are green due to the presence of chlorophyll, but the chloroplasts are not transmitted vertically (i.e., are absent in eggs and larvae) and do not undergo division in the sea slug [3].

Twenty adult specimens of *E. crispata* purchased from TMC Iberia (Lisbon, Portugal) were maintained in a 150 L recirculated life support system operated with artificial seawater at 25 °C, a salinity of 35 and an irradiance of 40 µmol photons m^−2^ s^−1^. These individuals were fed with the macroalga *Bryopsis plumosa* (Hudson) C. Agardh, 1823 and reproduced successfully, originating offspring of about 500 sea slugs. Following metamorphosis of the veliger larvae, the juveniles began feeding on the filamentous macroalga within 2–3 days after hatching, acquiring chloroplasts and turning green. However, one of the juveniles remained white and was unable to retain chloroplasts (Figure 2a,b). Chlorophyll fluorescence measurements with a fluorometer (Junior-PAM, Walz) showed the absence of minimum fluorescence (F_o_) of this white specimen, confirming the lack of chlorophyll. On the other hand, all other juvenile sea slugs showed both F_o_ and variable fluorescence (F_v_/F_m_ ≈ 0.73), indicating the presence of functional chloroplasts.

The “albino” specimen reached a maximum length of 7 mm, then stopped growing, and within 6 weeks post-metamorphosis it was significantly smaller than kleptoplast-bearing sea slugs (Figure 2c). The specimen was frozen for further analysis. It is possible that this aposymbiotic specimen immediately digested the chloroplasts together with the other algal cellular components, compromising its normal development. Pelletreau et al. [5] observed that the accumulation of photosynthesis-derived lipids in 1–14 days post-metamorphosis juveniles of *E. chlorotica* was necessary for chloroplast stability and establishment of permanent kleptoplasty. Translocation of photosynthates from kleptoplasts to the sacoglossan sea slug host has been shown to contribute to animal growth, survival and reproductive fitness [6,7,8].

Although the occurrence of less pigmented natural populations of photosynthetic sacoglossan sea slugs have been reported [9], we found no reference to an “albino” specimen as described here. Aposymbiotic specimens could be relevant in understanding the mechanisms involved in chloroplast recognition in Sacoglossa.

## Figures and Tables

**Figure 1 diversity-14-00313-f001:**
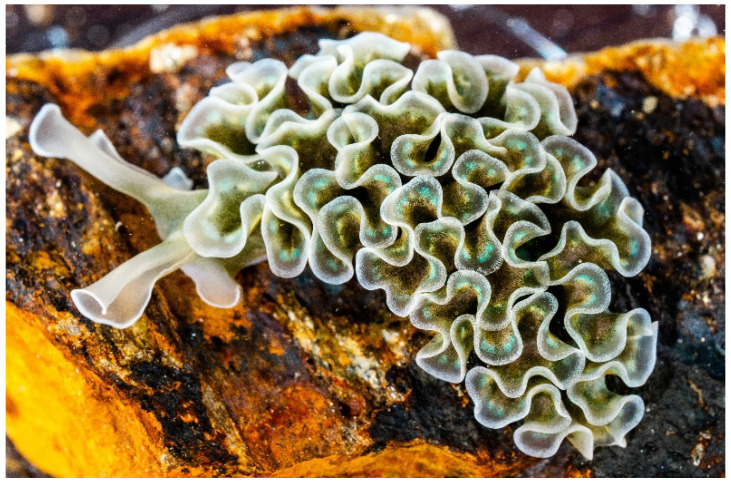
The sacoglossan sea slug *Elysia crispata* with its exuberant lateral folds (parapodia). The green coloration is due to the presence of chlorophyll within functional chloroplasts stolen from its algal food sources.

**Figure 2 diversity-14-00313-f002:**
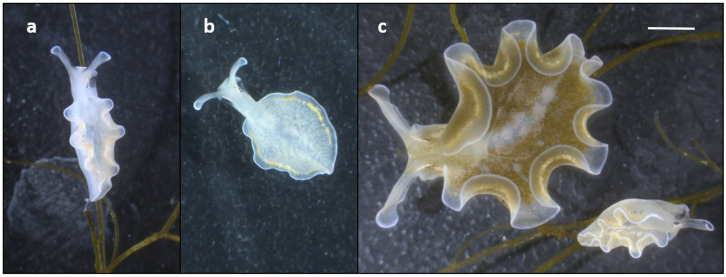
The aposymbiotic juvenile of *Elysia crispata* (**a**,**b**). The specimen was unable to retain chloroplasts from its food source, the macroalga *Bryopsis plumosa*. The “albino” sea slug was significantly smaller than its kleptoplast-bearing siblings (**c**). Chloroplast acquisition is required for the normal development of *E. crispata*. Scale bar: 2 mm.

## Data Availability

Not applicable.

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
