# Peer review of "Aposymbiotic Specimen of the Photosynthetic Sea Slug Elysia crispata"

_diversity, 2022, doi:10.3390/d14050313_

Round 1

Reviewer 1 Report

Sacoglossans are marine snails that live by feeding on algae, planting their chloroplasts in their body, exposing these chloroplasts to light and using the photosynthetic products of the chloroplasts for their growth. This brief paper describes a lab-hatched batch of sacoglossans in which one single individual failed to maintain chloroplasts; and consequently, he remained very small as compared with his brothers. The paper thus shows the importance of chloroplasts in the growth of saccoglossan snails. The paper is important, it is well written, and the figure is of cardinal importance. I suggest that this paper be accepted for publication, just as it is.

Author Response

I believe the option of “Extensive editing of English language and style required” selected by Reviewer #1 is an error as the Reviewer recommends publication as it is.

Reviewer 2 Report

This article reports the interesting observations of an albino specimens belonging to E. crispate sacoglossan species. The text is clear and concise and gives some useful insights on this sea slug species. I have just one point to raise to the author. It could be very useful to add pictures of the eggs and of the same specimens once adult, but I do not know if this is possible or if the authors have some photographs of the adult specimens.

I know that the specimens have been bought but, is it possible by chance to have information on the origin of the specimens purchased?

Apart from these curiosities, I recommend the publication of this interesting work only after having addressed the small suggestions below.

Line 9

Please write the full name of the species if it is at the beginning of the sentence.

lines 12-13

This last sentence needs to be linked to the rest of the abstract. For example, you could add ‘This evidence highlights that Chloroplast acquisition is required for the normal development of E. crispata.’

Line 16

Please add the author and year of the species.

Line 25

Is the green coloration due to the presence of functional chloroplasts or is it due to the presence of chlorophyl inside the chloroplasts?

Line 29

Please add author and year for the algal species Bryopsis plumosa.

Line 33

Please change ‘Fig. 2a-b’ with ‘Figs 2a-b’.

Lines 40-42 and 43-45

I personally do not like that figure caption is exactly identical to the text, so I kindly suggest to change figure caption to avoid repetition.

Line 49

Abbreviate Elysia in E.

Best regards.

Author Response

Reviewer #2

This article reports the interesting observations of an albino specimens belonging to E. crispate sacoglossan species. The text is clear and concise and gives some useful insights on this sea slug species. I have just one point to raise to the author. It could be very useful to add pictures of the eggs and of the same specimens once adult, but I do not know if this is possible or if the authors have some photographs of the adult specimens.

R: We do have photographs of the egg masses, individual eggs and larvae, as we are working on a paper on the early development of Elysia crispata. We believe it is out of the range of this small note.

I know that the specimens have been bought but, is it possible by chance to have information on the origin of the specimens purchased?

R: We did ask the supplier, but we did not obtain additional information on the animals.

Apart from these curiosities, I recommend the publication of this interesting work only after having addressed the small suggestions below.

Line 9

Please write the full name of the species if it is at the beginning of the sentence.

R: Corrected as suggested.

lines 12-13

This last sentence needs to be linked to the rest of the abstract. For example, you could add ‘This evidence highlights that Chloroplast acquisition is required for the normal development of E. crispata.’

R: Corrected as suggested.

Line 16

Please add the author and year of the species.

R: Information added as suggested.

Line 25

Is the green coloration due to the presence of functional chloroplasts or is it due to the presence of chlorophyl inside the chloroplasts?

R: To be exact from the chlorophyll inside the chloroplasts. The legend was changed for accuracy.

Line 29

Please add author and year for the algal species Bryopsis plumosa.

R: Information added as suggested.

Line 33

Please change ‘Fig. 2a-b’ with ‘Figs 2a-b’.

R: Corrected as suggested.

Lines 40-42 and 43-45

I personally do not like that figure caption is exactly identical to the text, so I kindly suggest to change figure caption to avoid repetition.

R: The figure legend was shortened to avoid repetition with the text.

Line 49

Abbreviate Elysia in E.

R: Abbreviated as suggested.

Reviewer 3 Report

No comments for the authors.

Author Response

Not applicable.